# In-Field Observation of Root Growth and Nitrogen Uptake Efficiency of Winter Oilseed Rape

**Julien Louvieaux** [1,2,*]**, Antoine Leclercq** [2]**, Loïc Haelterman** [1] **and Christian Hermans** [1]

[1]   Crop Production and Biostimulation Laboratory, Interfacultary School of Bioengineers,
    Université libre de Bruxelles, Campus Plaine CP 245, Bd du Triomphe, 1050 Brussels, Belgium;
    loic.haelterman@ulb.ac.be (L.H.); Christian.Hermans@ulb.ac.be (C.H.)

[2]   Laboratory of Applied Plant Ecophysiology, Haute Ecole Provinciale de Hainaut Condorcet, Centre pour
    l'Agronomie et l'Agro-industrie de la Province de Hainaut, 11 rue Paul Pastur, 7800 Ath, Belgium;
    a.leclercq@carah.be

*   Correspondence: julien.louvieaux@condorcet.be

**Abstract:** Field trials were conducted with two nitrogen applications (0 or 240 kg N ha$^{-1}$) and three modern cultivars of winter oilseed rape (*Brassica napus* L.) previously selected from a root morphology screen at a young developmental stage. The purpose is to examine the relationship between root morphology and Nitrogen Uptake Efficiency (NUpE) and to test the predictiveness of some canopy optical indices for seed quality and yield. A tube-rhizotron system was used to incorporate below-ground root growth information. Practically, clear tubes of one meter in length were installed in soil at an angle of 45°. The root development was followed with a camera at key growth stages in autumn (leaf development) and spring (stem elongation and flowering). Autumn was a critical time window to observe the root development, and exploration in deeper horizons (36–48 cm) was faster without any fertilization treatment. Analysis of the rhizotron images was challenging and it was not possible to clearly discriminate between cultivars. Canopy reflectance and leaf optical indices were measured with proximal sensors. The Normalized Difference Vegetation Index (NDVI) was a positive indicator of biomass and seed yield while the Nitrogen Balance Index (NBI) was a positive indicator of above-ground biomass N concentration at flowering and seed N concentration at harvest.

**Keywords:** nitrogen fertilization; nitrogen uptake efficiency; minirhizotron; oilseed rape; root development

## 1. Introduction

By the year 2050, a societal challenge will be to almost double the food production from existing land areas to feed more than an estimated nine billion people [1,2]. Innovation has therefore to make a step change to achieve environmentally sustainable intensification of agricultural systems. Crop production heavily relies on chemical fertilizers, but a considerable fraction of these gets lost with harmful consequences to the environment. Nitrogen (N) is the macronutrient required in the largest amount by crops. In agricultural ecosystems, N is continually depleted by such processes as exportation of N-containing crop residues, ammonia volatilization, microbial denitrification, erosion or N leaching. Therefore, the soil N reserve must be replenished every so often to sustain crop yield but N fertilizer in excess is detrimental to the human health (methemoglobinemia) and to the environment (eutrophication, greenhouse gas emission) [3,4]. One way to reduce N fertilizer input is to breed crops with greater Nitrogen Use Efficiency (NUE) [5,6]. NUE has two main components: the Nitrogen Uptake Efficiency (NUpE), which describes the plant capacity to acquire N from the soil, and the Nitrogen Utilization Efficiency (NUtE), the capacity to utilize N to produce harvestable organs [6].

Rapeseed (*Brassica napus* L.) is the second most important oilseed crop in the world [7]. Winter oilseed rape is regarded as a cash crop that diversifies cereal-dominated rotations and that reduces N leaching during autumn and winter [8]. While most of the efforts have been on proteolysis and N remobilization to the seeds [9,10], the root system is still considered as a "black box." Nonetheless, the root organ is central for N acquisition [11] and its morphology could be optimized [12]. The influence of nitrate on biomass allocation and root morphology is well documented in the model and parented plant *Arabidopsis thaliana*. During homogeneous low nitrate conditions, root growth is favored overshoot resulting into increased root to shoot biomass ratio [13], and the elongation of main and lateral roots is triggered [14–16]. A rationale shared by several authors is that a branched root system that explores a larger soil volume prevents N leaching [17–19]. There are indications that at low N supply, the seed yield of rapeseed cultivars is most closely correlated with N uptake (Nupt) [20] and root growth following stem extension [21]. Other reports indicate that N- efficient cultivars are characterized by an important root density during the vegetative growth stage [22]. Therefore, optimizing the root morphology may be a valuable strategy to improve NUpE. Furthermore, genetic differences in NUpE underpin variation in NUE under limiting N conditions [20,23]. This suggests that NUpE may be a valuable target for the creation of N-efficient genotypes.

In this study, a tube-rhizotron system was employed to characterize the spatiotemporal dynamics of the oilseed rape root system. Our aim is to observe the crop root growth at different key developmental stages and to evaluate such experimental setting for discriminating between N fertilization treatments and genotypes. We also tested whether one cultivar with a branched and deep root system would have a greater NUpE. Complementarily, proximal sensors based on canopy optical properties (foliar reflectance and pigment fluorescence) were evaluated.

## 2. Material and Methods

### 2.1. Plant Material

The three varieties of winter oilseed rape (*Brassica napus* L.) are restored hybrids registered in the French catalogue of plant species and varieties with the following names: DK Exception (DKE), DK Expansion (DKA), and Dualis (DUA). Seeds were obtained from the Terres Inovia network for post-registration evaluation of winter oilseed rape cultivars in Northern France. The plant development stages were determined using BBCH-identification keys of oilseed rape [24,25].

### 2.2. Field Site and Experimental Design

Field trials were conducted from September 2017 to July 2018 at the CARAH experimental farm in Ath, Belgium (N 50°36′52.038″; E 3°45′54.025″), on a Luvisol soil (>2 m depth) with favorable natural drainage. Preceding crop was winter wheat. At sowing, N mineral content in 0–30 cm soil horizon was 78.6 kg ha$^{-1}$. The experimental design consisted of split-plot randomized blocks with four replicates. On 28 August 2017, microplots (1.5 m × 12 m) were sown at a density of 60 seeds m$^{-2}$. Two nitrogen fertilization treatments were considered: control/unfertilized plots (N−) and N fertilized plots with 240 kg ha$^{-1}$ (N+). In the latter case, two doses of ammonium nitrate were applied: 60 kg N ha$^{-1}$ at BBCH 00 and 180 kg N ha$^{-1}$ at BBCH 31–32. The purpose of applying fertilizer at sowing was to observe the impact on root development during crop establishment in autumn. Temperature data were obtained from a meteorological station located less than 600 m from the field. Accumulated average daily temperature corresponding to growing degree days was calculated assuming a base temperature of 4.5 °C (GDD$_{4.5}$) [26].

### 2.3. Root Scanning and Data Analysis

Root system expansion in the microplots was monitored with a minirhizotron system (CI-600, CID Bio-Science, Camas, USA). Immediately after sowing, transparent plexiglass tubes (105 cm length, 7 cm external diameter) were inserted with a manual auger of the same diameter (Eijkelkamp, Giesbeek,

The Netherlands) at an angle of 45° [27]. A total of 18 tubes were installed, with 6 tubes per cultivars and 3 tubes per N treatment. The above-ground part of the tubes (15–20 cm) was covered with dark plastic to block out light. The minirhizotrons reached a depth of approximatively 0.5 m, corresponding to the upper soil layer in which fertilizers are most likely intercepted by roots. The captured images of the roots around the tube outer surface were divided in four zones (A, B, C, D) corresponding to soil horizons with an increasing depth of 12 cm, from 0 cm (ground level) to 48 cm depth (given the angle of the tube at 45° off the vertical) (Figure 1).

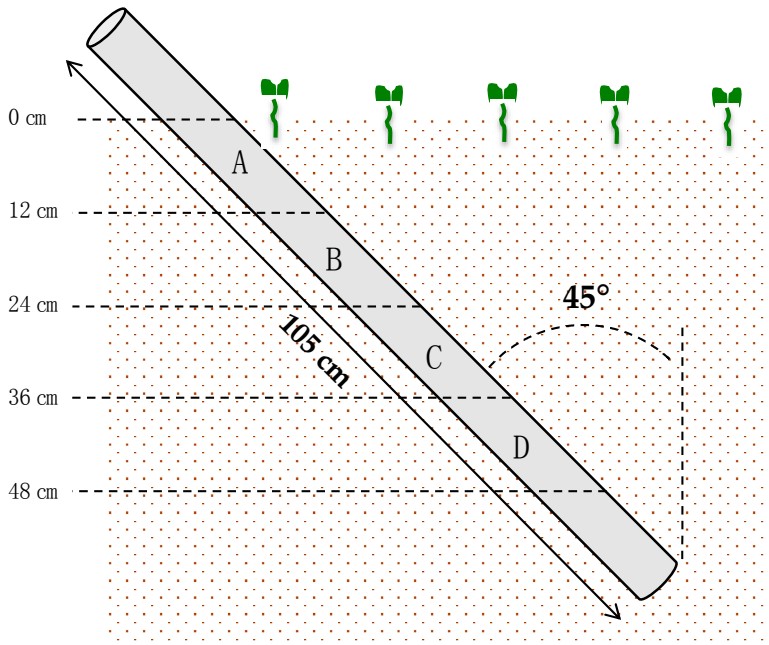

**Figure 1.** Minirhizotron positioning in soil. Transparent tubes (105 cm length, 7 cm diameter) were inserted at 45° angle with a manual auger. Four observation zones were defined: A (0–12 cm), B (12–24 cm), C (24–36 cm) and D (36–48 cm).

Captures of the soil profiles were redrawn with a digital pen and images were then quantified using Optimas v6.1 (Media Cybernetics). Lines were identified with the "automatically find lines" feature. The lengths of individual lines were obtained through the measurement explorer dialog box and summed for each image. The root length surface density (RLSD) was calculated as the length of visible roots per surface (mm cm$^{-2}$), following to previous studies [28,29].

### 2.4. Determining Aerial Biomass, Nitrogen Concentration, Nitrogen Uptake, and Nitrogen Uptake Efficiency

Above-ground biomasses covering a plot surface of 0.75 m$^2$ were collected on 27 November 2017 (BBCH 19, 647 GDD$_{4.5}$) and 25 April 2018 (BBCH 65, 1026 GDD$_{4.5}$). Samples were dried for 48 h at 70 °C. Nitrogen concentration in shoot tissues (N%$_{shoot}$) was determined with a Vario MAX Cube (Elementar, Langenselbold, Germany). Nitrogen uptake (Nupt) was calculated as the product of biomass and N%s$_{hoot}$. Nitrogen Uptake Efficiency (NUpE) was calculated as follows [30]: NUpE = (Nupt$_{N+}$ − Nupt$_{N-}$)/N, where Nupt$_{N+}$ is N uptake after fertilization treatment and Nupt$_{N-}$ without fertilization treatment and $\Delta$N is the difference of applied nitrogen amount between N+ and N− treatments.

### 2.5. Canopy Optical Properties

The Normalized Difference Vegetation Index (NDVI) was measured on 30 October 2017 (BBCH 19) with the GreenSeeker® handheld crop sensor (Trimble, Sunnyvale, USA). The sensor was positioned at 60 cm from the top of the canopy to measure the reflectance at 660 nm (red) and 770 nm (near infrared) [31]. Ten measurements were made per microplot. The NDVI is commonly used as an

indicator of aerial biomass production and nutrient status of crops [32–34]. In canola, an optimal time window to measure NDVI for grain yield prediction is between 50 and 70 days after sowing (DAS), while later readings are poorly predictive because they are influenced by stem elongation and flowering [35,36]. In the present study, NDVI measurements were made at 63 DAS (BBCH 19).

The chlorophyll index (CHL), flavonol index (FLAV), and the Nitrogen Balance Index (NBI) were measured on 23 April 2018 (BBCH 65) with the Dualex® Scientific + (Force-A, Orsay, France), a leafclip sensor. Within each microplot, measurements were conducted on the mature leaves of five plants.

## 2.6. Yield and Harvest Quality Traits

The central parts of each plot, corresponding to a surface of 10.5 m², were harvested with a combine harvester (Wintersteiger Delta, Ried im Innkreis, Austria). Subsamples of seeds were analyzed for humidity, OilConc, ProtConc, and GLS (Table 1) by near infra-red spectroscopy (XDS NIR analyzer, Foss, Hilleroed, Denmark). A seed counter was used to determine TSW (Numigral, Chopin Technologies, Villeneuve-la-Garenne, France).

**Table 1.** Definition of yield and harvest quality parameters.

| Abbreviation. | Description |
|---|---|
| SY | Seed yield corrected to a standard water content of 9% |
| TSW | 1000 seed weight with moisture adjusted to 9% |
| ProtConc | Protein concentration in dry seeds |
| NConc | Nitrogen concentration in dry seeds = ProtConc/6.25 |
| OilConc | Oil concentration in dry seeds |
| SNU | Seed nitrogen uptake = NConc × SY |
| ProteinY | Protein yield = ProtConc × SY |
| OilY | Oil yield = OilConc × SY |
| GLS | Glucosinolate concentration in dry seeds |

## 2.7. Statistical Analysis

The split-plot experimental design and the data statistical analyses were done with Statbox v7.0 (FBC Software). Assumptions for analysis of variance (ANOVA) were verified with D'Agostino-Spearman test. During ANOVA, the following variances were calculated: cultivar, fertilization, interaction (cultivar × fertilization), blocks (controlled factor), sub-blocks (controlled factor), and Residuals. ANOVA and Pearson's correlations were performed at a significance level of $\alpha$ = 0.05. Newman-Keuls post-hoc tests were employed to classify the different results into homogeneous groups with different letters corresponding to homogeneous groups.

## 3. Results

### 3.1. Selecting Three Cultivars with Contrasting Root Morphologies

Three winter oilseed rape cultivars with dissimilar root morphologies were identified during a previous laboratory screening [37]. Twelve seedlings of DK Exception (DKE), DK Expansion (DKA), and Dualis (DUA) grew for one week in hydroponics. Cultivars were ranked in order of increasing root biomass production and lateral root length (Figure S1, Table S1). Eventually, these three genotypes were cultivated in field with two N fertilization treatments (N− or N+).

### 3.2. Observing Root Development Below Ground with a Camera System

Clear tubes were installed between plant rows to observe root development. The roots on the outer surface of the rhizotron (Figure 2) were imaged with a rotating scanner at key developmental stages over the seasons: crop establishment (BBCH 15, 18, and 19) during autumn, as well as stem elongation (BBCH 55) and flowering (BBCH 65) during spring.

The root system progressively explored soil layers and growth was the fastest at the end of autumn. After winter, roots were thinner and less visible. A first comparison was made between the two N fertilization treatments based on average values of the three genotypes. At BBCH 18, root length surface density (RLSD) in the deep soil horizon (36–48 cm) was significantly ($P < 0.05$) more important in N− compared to N+ conditions (Figure 3b). This clearly indicated a stimulation of root growth in the absence of N fertilization. A second comparison was done between cultivars in response to N treatment. There was an important heterogeneity between rhizotron images, but no obvious difference in rooting depth and placement was observed between cultivars across time points and N treatments (Figure 3f–j).

### 3.3. Above-Ground Biomass Production and Nitrogen Uptake Efficiency

Biomass and nitrogen concentration in plant tissues were measured following N treatments to calculate NUpE. At BBCH 19, the above-ground dry biomass was 56.0% more important in N+ compared to N− plots and the N concentration was 29.7% greater (Table 2). The total N uptake was approximately double between treatments. Therefore, the NUpE value was close to one, and this indicated the entire N fertilizer amount (60 kg N ha$^{-1}$) applied in N+ at sowing was absorbed by the culture by autumn end. The NUpE differed between genotypes but not significantly ($P > 0.05$). At BBCH65, the above-ground dry biomass only increased by +5.9% in N+ compared to N− plots and the N content increased by +27.8%. The calculated NUpE was 0.24, suggesting that the second N fertilizer application was not totally taken up at this developmental stage. Under N+ treatment, the cultivars DKA and DUA showed significant ($P < 0.05$) greater Nupt compared to DKE (+21.0% and +27.4% respectively), those two cultivars having also the most branched root system in the laboratory screen (Figure S1).

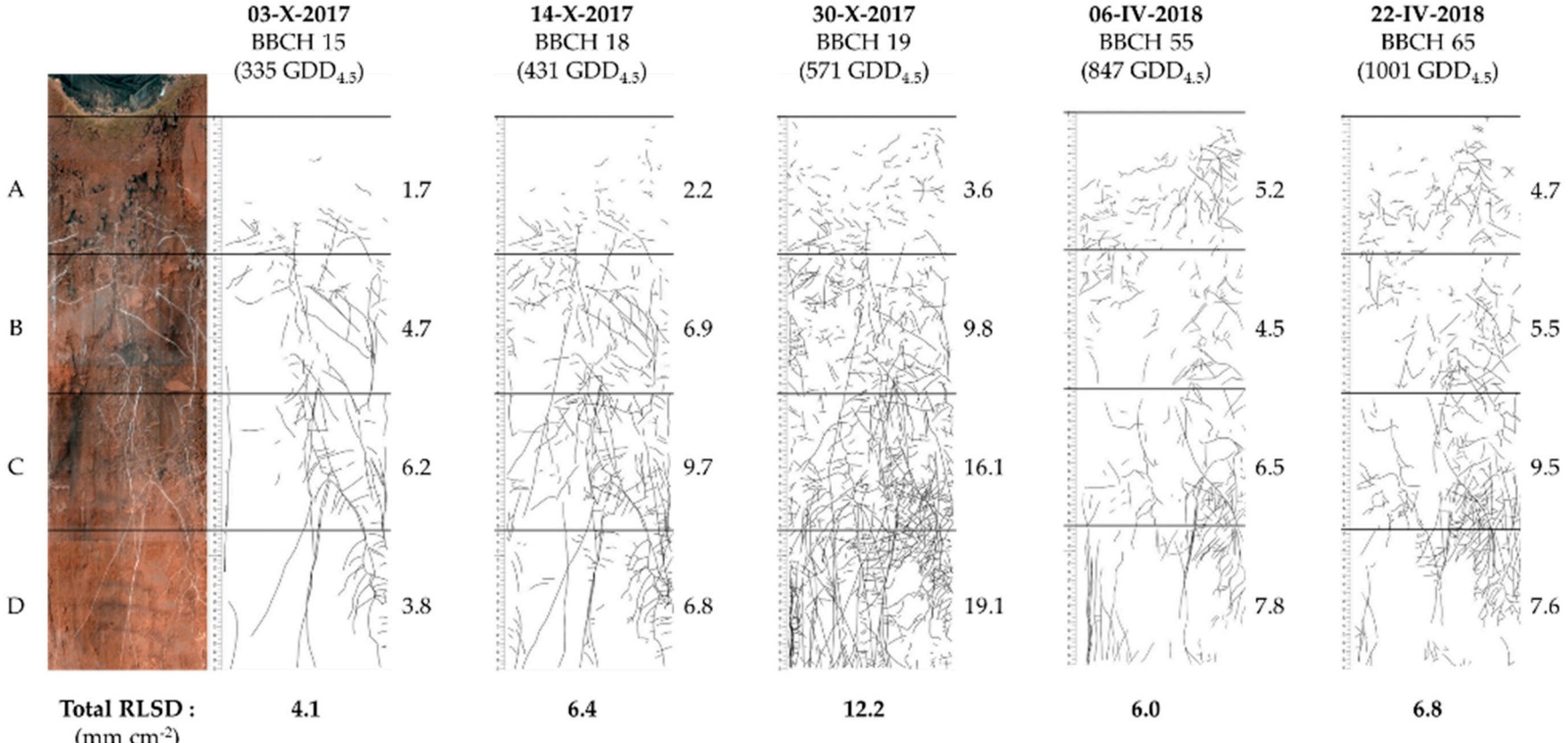

**Figure 2.** Example of image analysis and quantification of root length surface density. The DK Exception (DKE) cultivar received N+ treatment. Root length surface density (RLSD) values (mm cm$^{-2}$) are shown in different soil horizons and at different time points of the growing season. The soil horizons are indicated by different letters: A = 0–12 cm; B = 12–24 cm; C = 24–36 cm, and D = 36–48 cm. The BBCH-scale indicates phenological development stages [24,25]. GDD: growing degree days.

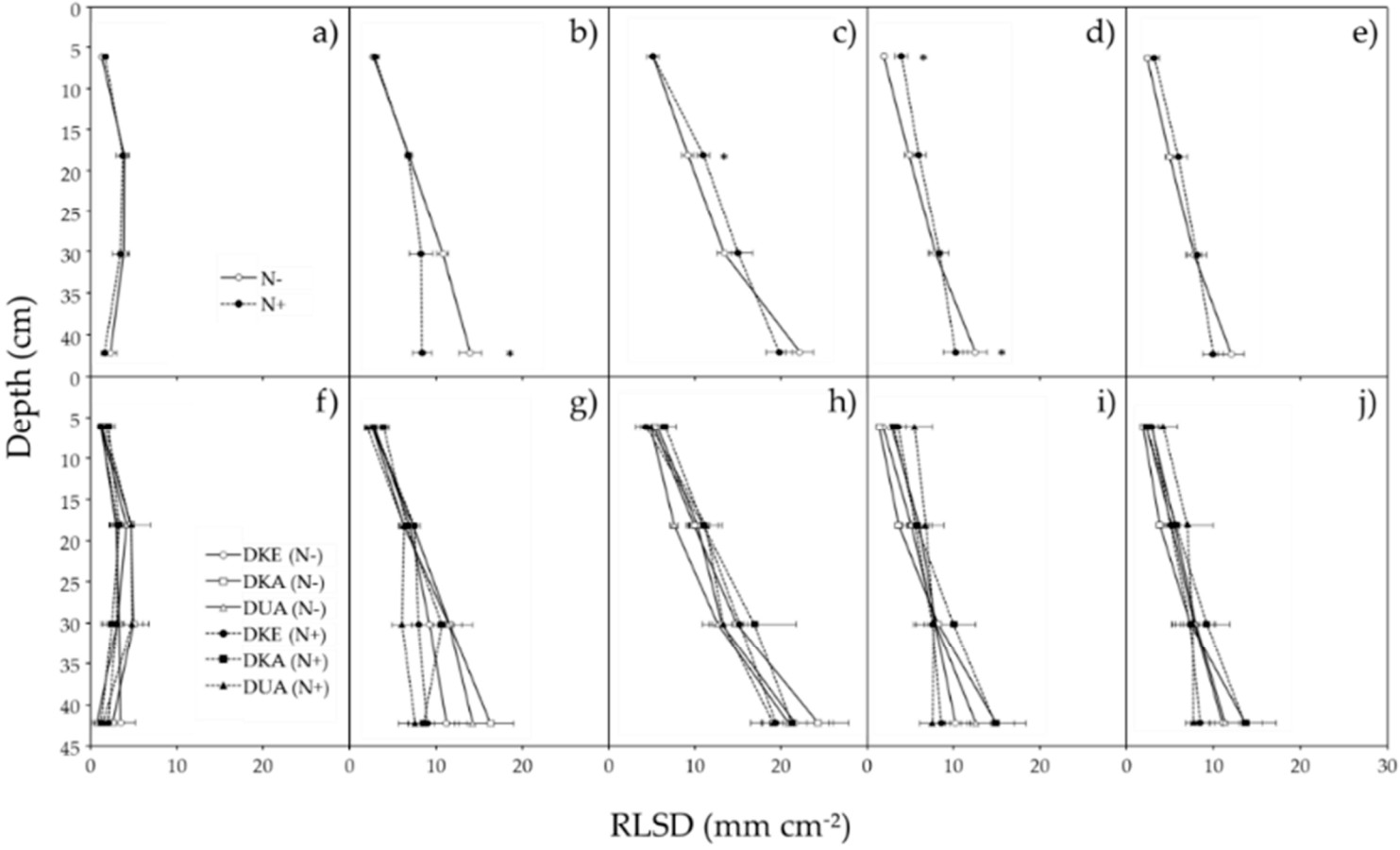

**Figure 3.** Root length surface density of three oilseed rape cultivars in different soil depths and at different time points of the growing season. Measurements were done at BBCH 15 (**a,f**), BBCH 18 (**b,g**), BBCH 19 (**c,h**), BBCH 55 (**d,i**), and BBCH 65 (**e,j**). Mean results according to the fertilizer treatments (n = 9 ± s.e.) (**a–e**) and to individual cultivars (n = 3 ± s.e) (**f–j**). Stars indicate significant differences at level $\alpha$ = 0.05.

**Table 2.** Above-ground biomass measurements and harvest parameters. Mean values according to the fertilizer treatments (n = 12 ± s.e.) and to individual cultivars (n = 4 ± s.e.). Letters indicate statistical differences at significance level α = 0.05. Pooled standard errors (s.e.$_{pooled}$) of N− and N+ populations are mentioned according to fertilizer treatments and cultivars. The effects of different factors as a percentage of the observed variation are given in Figure S2.

| | | BBCH 19 | | | | BBCH 65 | | | |
|---|---|---|---|---|---|---|---|---|---|
| | | **Dry Biomass** | **N%$_{shoot}$** | **Nupt** | **NUpE** | **Dry Biomass** | **N%$_{shoot}$** | **Nupt** | **NUpE** |
| | | **(t ha$^{-1}$)** | | **(kg ha$^{-1}$)** | | **(t ha$^{-1}$)** | | **(kg ha$^{-1}$)** | |
| **Mean values** | N− | 2.31 [b] ± 0.17 | 2.63 [b] ± 0.09 | 61.4 [b] ± 5.6 | | 6.75 ± 0.42 | 2.50 [b] ± 0.10 | 168.5 [b] ± 11.7 | |
| | N+ | 3.60 [a] ± 0.14 | 3.41 [a] ± 0.11 | 122.2 [a] ± 5.0 | 1.01 ± 0.11 | 7.14 ± 0.39 | 3.20 [a] ± 0.08 | 227.1 [a] ± 11.8 | 0.24 ± 0.06 |
| | s.e.$_{pooled}$ | ±0.22 | ±0.14 | ±7.4 | | ±0.57 | ±0.12 | ±16.6 | |
| **DKE** | N− | 2.66 ± 0.38 | 2.50 ± 0.17 | 67.9 ± 12.6 | | 5.75 ± 0.76 | 2.53 ± 0.18 | 144.3 ± 17.8 | |
| | N+ | 3.52 ± 0.14 | 3.42 ± 0.16 | 120.4 ± 7.7 | 0.88 ± 0.23 | 5.92 ± 0.52 | 3.34 ± 0.13 | 195.6 [b] ± 10.2 | 0.21 ± 0.08 |
| | s.e.$_{pooled}$ | ±0.41 | ±0.23 | ±14.7 | | ±0.92 | ±0.22 | ±20.5 | |
| **DKA** | N− | 1.92 ± 0.17 | 2.67 ± 0.15 | 51.8 ± 7.0 | | 7.45 ± 0.84 | 2.63 ± 0.17 | 194.8 ± 22.2 | |
| | N+ | 3.71 ± 0.28 | 3.36 ± 0.16 | 124.0 ± 8.3 | 1.20 ± 0.20 | 7.62 ± 0.46 | 3.10 ± 0.11 | 236.7 [a] ± 19.7 | 0.17 ± 0.14 |
| | s.e.$_{pooled}$ | ±0.33 | ±0.22 | ±10.8 | | ±0.96 | ±0.21 | ±27.9 | |
| **DUA** | N− | 2.35 ± 0.20 | 2.72 ± 0.18 | 64.6 ± 9.2 | | 7.04 ± 0.33 | 2.35 ± 0.17 | 166.4 ± 16.3 | |
| | N+ | 3.57 ± 0.34 | 3.45 ± 0.28 | 122.1 ± 11.9 | 0.96 ± 0.14 | 7.88 ± 0.65 | 3.16 ± 0.15 | 249.2 [a] ± 22.6 | 0.34 ± 0.13 |
| | s.e.$_{pooled}$ | ±0.39 | ±0.33 | ±15.0 | | ±0.73 | ±0.23 | ±27.9 | |

| | | Harvest Parameters | | | | | | |
|---|---|---|---|---|---|---|---|---|
| | | **SY** | **NConc** | **OilConc** | **GLS** | **TSW** | **SNU** | **NUpE** |
| | | **(t ha$^{-1}$)** | **(%)** | **(%)** | **(µmol g$^{-1}$)** | **(g)** | **(kg ha$^{-1}$)** | |
| **Mean values** | N− | 3.61 [b] ± 0.15 | 3.36 [b] ± 0.03 | 46.6 [a] ± 0.3 | 16.2 ± 0.4 | 5.16 ± 0.11 | 110 [b] ± 4 | |
| | N+ | 4.64 [a] ± 0.11 | 3.51 [a] ± 0.02 | 45.3 [b] ± 0.2 | 17.0 ± 0.5 | 5.15 ± 0.11 | 148 [a] ± 4 | 0.16 ± 0.02 |
| | s.e.$_{pooled}$ | ±0.18 | ±0.04 | ±0.3 | ±0.7 | ±0.15 | ±6 | |
| **DKE** | N− | 3.63 ± 0.34 | 3.40 ± 0.03 | 45.6 [bc] ± 0.2 | 17.2 ± 0.2 | 5.46 ± 0.10 | 112 ± 11 | |
| | N+ | 4.62 ± 0.21 | 3.50 ± 0.01 | 44.7 [c] ± 0.2 | 17.9 ± 1.1 | 5.44 ± 0.08 | 147 ± 6 | 0.14 ± 0.03 |
| | s.e.$_{pooled}$ | ±0.40 | ±0.04 | ±0.2 | ±1.1 | ±0.13 | ±13 | |
| **DKA** | N− | 3.40 ± 0.24 | 3.43 ± 0.05 | 46.4 [b] ± 0.4 | 16.4 ± 0.9 | 5.30 ± 0.06 | 106 ± 7 | |
| | N+ | 4.86 ± 0.13 | 3.55 ± 0.03 | 45.7 [bc] ± 0.1 | 17.3 ± 0.8 | 5.32 ± 0.08 | 157 ± 5 | 0.21 ± 0.02 |
| | s.e.$_{pooled}$ | ±0.27 | ±0.06 | ±0.4 | ±1.2 | ±0.10 | ±9 | |
| **DUA** | N− | 3.80 ± 0.20 | 3.24 ± 0.04 | 47.7 [a] ± 0.1 | 15.0 ± 0.4 | 4.71 ± 0.09 | 112 ± 5 | |
| | N+ | 4.43 ± 0.21 | 3.49 ± 0.03 | 45.6 [bc] ± 0.2 | 15.7 ± 0.5 | 4.69 ± 0.03 | 140 ± 5 | 0.12 ± 0.04 |
| | s.e.$_{pooled}$ | ±0.29 | ±0.05 | ±0.2 | ±0.7 | ±0.10 | ±8 | |

### 3.4. Complementary Canopy Measurements

Pre-winter development of aerial biomass drives oilseed rape yield formation [38]. The Normalized Difference Vegetation Index (NDVI) is commonly used as an indicator of aerial biomass production and nutrient status in crops [32–34]. We then employed a proximal optical sensor (Greeseeker) to non-invasively probe leaf reflectance to assess plant vitality. Currently, this technique is also used in precision agriculture with drones and models to adjust nitrogen fertilizer levels [39]. In canola, an optimal time window to measure NDVI for grain yield prediction is between 50 to 70 days after sowing (DAS), while later readings are poorly predictive because they are influenced by stem elongation and flowering [35,36]. In this study, NDVI was measured at 63 DAS (BBCH 19). Another optical sensor (Dualex® leafclip) was used at flowering (BBCH 65), when nutrient absorption is maximum and leaf reflectance unsuitable. The Nitrogen Balance Index (NBI) is measured by leaf pigment fluorescence and is an indicator of plant N status [40–42].

At BBCH 19, the NDVI increased by +4.5% in N+ compared to N− plots. The cultivar DUA had significant ($P < 0.05$) greater values than DKE and DKA at both N treatments (Table 3). At BBCH 65, the NBI increased by +15.8% in N+ compared to N− plots. That parameter was greatly responding to N fertilization, but no difference was observed between cultivars.

**Table 3.** Leaf optical parameters in response to nitrogen fertilization treatment. The Normalized Difference Vegetation Index (NDVI) and the Nitrogen Balance Index (NBI) were measured on canopy of control (N−) or fertilized (N+) plots respectively at BBCH 19 and BBCH 65. Mean values according to the fertilizer treatments (n = 12 ± std) and to individual cultivars (n = 4 ± std). Pooled standard deviations ($std_{pooled}$) of N− and N+ populations are mentioned according to the fertilizer treatment and cultivars. Letters indicate statistical differences at significance level $\alpha = 0.05$.

|  |  | NDVI | NBI |
|---|---|---|---|
|  |  | **(BBCH 19)** | **(BBCH 65)** |
| **Mean values** | N− | 0.80 [b] ± 0.03 | 29.8 [b] ± 2.5 |
|  | N+ | 0.84 [a] ± 0.01 | 34.6 [a] ± 3.9 |
|  | $std_{pooled}$ | ±0.02 | ±3.3 |
| **DKE** | N− | 0.79 ± 0.03 | 29.4 ± 0.8 |
|  | N+ | 0.84 ± 0.01 | 36.0 ± 5.5 |
|  | $std_{pooled}$ | ±0.02 | ±3.9 |
| **DKA** | N− | 0.80 ± 0.03 | 31.3 ± 3.9 |
|  | N+ | 0.84 ± 0.01 | 36.0 ± 2.6 |
|  | $std_{pooled}$ | ±0.02 | ±3.3 |
| **DUA** | N− | 0.82 ± 0.03 | 28.8 ± 1.7 |
|  | N+ | 0.85 ± 0.01 | 31.7 ± 1.8 |
|  | $std_{pooled}$ | ±0.02 | ±1.7 |

### 3.5. Yield and Harvest Quality

Upon harvest, yield and seed quality traits were measured (Table 1). SY increased by 28.4% in N+ compared to N− plots (4.64 ± 0.39 vs. 3.61 ± 0.51 t ha$^{-1}$, respectively) and the NConc increased by 4.7% (3.51 ± 0.06 vs. 3.36 ± 0.11 %) (Figure 4, Table 2). NConc and OilConc were negatively correlated, confirming previous findings [43,44]. The GLS and TSW were essentially cultivar dependent.

NUpE was estimated at harvest, considering the total nitrogen absorbed by harvestable organs (SNU). No marked difference was observed, and values were lower than the one observed at flowering (Table 2), reflecting a poor ability to remobilize nitrogen to harvest organs.

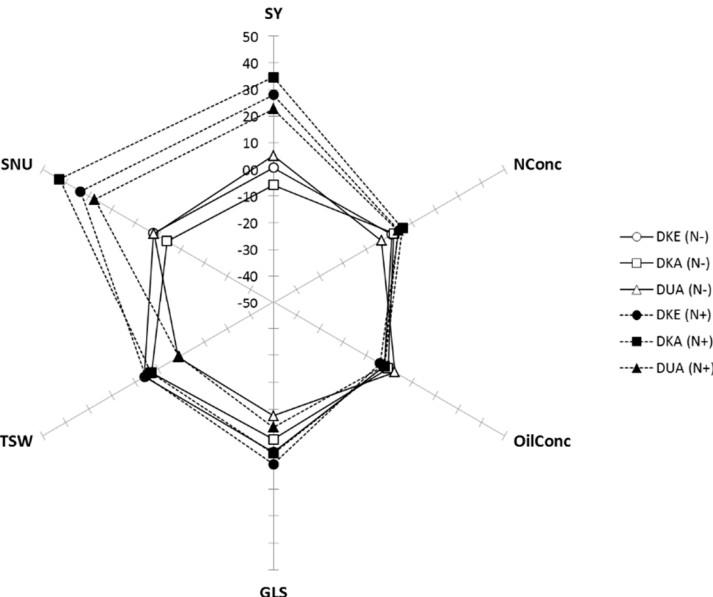

**Figure 4.** Comparison of yield and harvest quality parameters. The spider plot shows the percentage variation of the normalized trait values relative to the average of the unfertilized plots (0% indicates no difference compared to the average of N− plots). Parameters are defined in Table 1.

### 3.6. Predictiveness of Optical Parameters

In order to evaluate the predictiveness of the biosensing methods, indices measured at the canopy level during vegetative growth and flowering stages were correlated with yield and harvest quality parameters (Table S2). At the end of autumn (BBCH 19), NDVI and above-ground dry biomass were positively correlated (r = 0.83), and both negatively with RLSD (r = −0.60; −0.53 respectively). At full flowering (BBCH 65), NBI was only positively correlated (r = 0.67) with N concentration in above-ground plant biomass (N%$_{shoot}$).

Furthermore, some canopy measurements recorded during vegetative growth in autumn and at flowering were significantly correlated to harvest parameters. The NDVI readings in autumn were correlated to SY (r = 0.75), while NBI at flowering to NConc in seeds and SNU (r = 0.59 and r = 0.47, respectively).

### 4. Discussion

Improving NUE is essential to ensure the environmental and economic sustainability of oilseed rape production. While the aerial organs are extensively considered in breeding programs, equivalent examinations in roots are receiving much less attention because of the practical challenge of working with the below-ground organs [45]. In situ observation of the root system is not trivial and conventional methods (e.g., soil coring) can be tedious [46]. Here, we employed a minirhizotron system to observe dynamic aspects of root growth with a camera through clear tubes inserted in soil (Figure 1). That non-destructive method allows following root development in soil horizons during the culture cycle with a fair imaging resolution. Creating an intimate contact between transparent tube and soil improves the image quality [27], notwithstanding soil texture and moisture can hamper the further image analysis process. In this study, image processing with the dedicated manufacturer software was not satisfactory enough because of the root overlapping and water condensation outside the tubes, as also reported in other studies [47]. To overcome that problem, root items were redrawn by hand, but it was time consuming. Current tools for processing rhizotron images are often restricted to a limited number of variables and show limitation with highly branched root systems [48]. Improvement has still to be done to take full advantage of the high throughput scanning method. Before wintertime, we observed the full deployment of the root system (Figures 2 and 3). The end of autumn season was a critical

time window to observe root development. At BBCH 18, N fertilization treatment reduced RLSD in the deep soil horizon (Figure 3) and stimulated above-ground biomass production (Table 1). During spring, roots were less clearly visible, resulting in lower RLSD values (Figures 2 and 3). This could be attributed to less important resource allocation to the roots, to frost damage during winter, and/or to deeper soil exploration, out of sight of the imaging system (>0.5 m). Indeed, the root system of oilseed rape can reach more than 2.3 m depth, with roots up to 1.5 m before winter [30].

Winter oilseed rape is seen as a poor N− efficient crop that has a low seed production per N unit applied, mainly because of a low N remobilization from vegetative to reproductive organs [39,49]. Nonetheless, the plant can absorb great N amounts during autumn and winter, making it a valuable catch crop to reduce N leaching in soil [50]. Assessing NUpE at the beginning of the growth cycle is crucial, since up to one-third of the total N uptake occurs before winter [39]. The NUpE values were greater in autumn than in spring (Table 2). This can be explained by a fraction of N absorbed before winter being lost with littering of frozen leaves, but still susceptible of being mineralized and reabsorbed during spring [49,51]. An estimation of that lost is between 20 and 100 kg N ha$^{-1}$ [51,52]. During stem elongation and onwards, occurs the sequential fall of senescent leaves with large N content (2–2.5% of dry weight) [49]. The pod filling is an important phase during which plants are still actively taking up N [39,49]. The sink capacity of the pods is considered as a bottleneck for N remobilization and this could be enhanced through yield improvement [53]. At harvest, we estimated nearly one-fifth of applied N was recovered in the seeds (Table 2), that proportion being lower than in previous reports [49,52,54]. Large N input (240 kg ha$^{-1}$) and drought stress likely contributed to this low value. Indeed, the year 2018 weather conditions were extreme, and two-thirds less precipitation than annual average during flowering and pod filling, severely depressed yield. Furthermore, the negative impact of drought stress on N recovery to seeds is well documented [52,55].

We appraised two rapid and non-destructive proximal remote sensing methods based on optical properties of plants. The NDVI at the canopy level during autumn was a good estimate of above-ground biomass production in autumn and seed yield at harvest, while the NBI at the leaf level during flowering time, of N concentration in plants tissues and seeds at harvest. Previous studies showed that NUE in oilseed rape was strongly correlated to leaf N content at flowering, especially under low N supply [56]. Therefore, NBI measurements could be convenient for improving rapid screening procedures. The NDVI and NBI delivered complementary information but they were not correlated to each other.

We based our research program on the premise that NUpE could be improved by favoring a more branched root system that explores a larger soil volume to limit N leaching [37]. In a previous report, we challenged a panel of winter oilseed rape cultivars at a young developmental stage (one-week-old seedlings) with divergent nitrate supplies and identified contrasting root morphologies [37]. Some roots traits of young seedlings (e.g., length of the primary root length) were positively correlated to NConc in subsequent field trials. Such a screening strategy in laboratory conditions may speed up for breeders the delivery of genotypes with ideal morphological features. Positive correlation between root traits in laboratory conditions and field performance was also observed in other studies [57]. For example, the primary root length measured in hydroponics was positively correlated to the field yield, and the lateral root density to the concentration of mineral elements in leaves. In this paper, in-field differences between cultivars were not marked regardless of the N fertilization treatments. Nonetheless, at BBCH 65 and following N+ treatment, the DKA and DUA cultivars had greater Nupt than DKE, the latter one showed less lateral root outgrowth in laboratory conditions (Figure S1, Table S1).

## 5. Conclusions

We have carried a one-year pilot experiment with three hybrid cultivars reflecting the market trends. Hybrids are high-yielding lines with enhanced NUE [58]. In perspective, more genotypes with contrasting root morphologies could be screened to further strengthen the corroboration between root morphological features in laboratory and soil conditions.

**Supplementary Materials:** The following are available online at http://www.mdpi.com/2073-4395/10/1/105/s1, Figure S1: Root morphologies of three oilseed rape cultivars cultivated in hydroponics, Figure S2: Global ANOVA of measured traits at different periods in field experiment, Table S1: Root system parameters of three oilseed rape cultivars cultivated in hydroponics, Table S2: Pearson's correlation coefficients between parameters measured at different time points.

**Author Contributions:** Conceptualization, J.L. and C.H.; formal analysis, J.L. and A.L.; funding acquisition, J.L. and C.H.; investigation, J.L. and A.L.; methodology, J.L. and L.H.; writing—original draft, J.L.; writing—review and editing, J.L., L.H., and C.H. All authors have read and agreed to the published version of the manuscript.

**Funding:** This research was funded by MIS and PDR T.0116.19 grants from Fonds de la Recherche Scientifique (F.R.S.-FNRS).

**Acknowledgments:** C.H. is an F.R.S.-FNRS research associate. The authors thankfully acknowledge the Centre pour l'Agronomie et l'Agro-industrie de la Province de Hainaut (CARAH) for the field trial and analyses.

**Conflicts of Interest:** The authors declare no conflict of interest.

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
