# Peer review of "In-Field Observation of Root Growth and Nitrogen Uptake Efficiency of Winter Oilseed Rape"

_agronomy, doi:10.3390/agronomy10010105_

Round 1

Reviewer 1 Report

The manuscript “In-field observation of root growth and nitrogen uptake efficiency of winter oilseed rape” reports interesting data about interaction between genotype and fertiliser in order to increase NUE.

Deep analysis of the rooting system is reported.

In general, the paper is well written and clear, in a sufficient English quality.

Some minor aspects would ask for some improvements, mainly in the material and methods section:

First: the informatics process used for determining root parameters are not well described. Specific information about the software used would improve the understanding of the paper.

Second: information about the equipment used to calculate NDVI and the other reflectance indexes are not reported. Specific indexes can vary based on the specific bands used for determination as they can vary in specific range.

Finally: Statistical analysis is quite lacking. Aussumptions have not been checked and results lacks of information about sample variability as only goup variability are reported, but not pooled variance estimates.

Some more specific issues:

Line 33: Maybe, in general runoff does not represent the most important term in the N cycle. Other, such as leaching and ammonia volatilisation are more important.

Line 112: not clear which other optical sensors were used.

Author Response

Thank you for your review. Please find our responses in attachment.

Best regards.

Reviewer 2 Report

Dear editors, dear authors,

The manuscript entitled “In-field observation of root growth and nitrogen uptake efficiency of winter oilseed rape” is about observing crop root growth at different key developmental stages and evaluating such experimental setting for discriminating between N fertilization treatments and oilseed rape genotypes.

General comments:

In general, the article is comprehensive and the topic is nicely presented.

My main concerns and suggestions for improvement are:

Please find some detailed comments on the text below:

Abstract:

Line 11: isn’t it (0 and 240 kg N ha-1)?

Introduction

Line 33:  Crop  production  heavily  relies  on  chemical  fertilizers,  but  a  considerable  fraction  gets  lost  due to leaching and erosion.

Line 35: processes

Line 36: denitrification and N leaching.

Line 37: N instead of nitrate

Line 60: profuse root systems refers to what?  Profuse rooting depth, root biomass, or root length density?

What are main effects of N limitation and N excess to root systems (in general an of oilseed rape)?

Please cite some literature.

There is some literature on oilseed rape root growth see, f.i. “Comparing Macropore Exploration by Faba Bean, Wheat, Barley and Oilseed Rape Roots Using In Situ Endoscopy”  DOI:  10.1007/s42729-019-00069-0  and other work e.g. from Eusun Han

Results

Isn’t it contrasting root morphologies

2.1 how many plants (per cultivar) were tested? Replicates? Please add here or in M&M section

Line 75 What is soil strata? Soil layers?

Line 81 you report of the large heterogeneity -> how many replicates (not mentioned in M&M)??

Line 88 Do you mean “. At BBCH 19, the above-ground dry biomass was 56.0 % higher in N+ compared to…”?

Line 89/90: you mean in treatment N+? Or both treatments?

Line 91 “No significant differences were observed between genotypes.” Refers to what? NUpE  differs but not significantly?

Methods

Please add the city and company to all devices, e.g.  instrument x (city, company)

line 210 one . too much after 4.1.

Do you know the initial Nmin concentration of the soil at sowing?

Line 223 soil type? Horizon depths?

Line 251 is aerial biomass above-ground biomass?

Line 254 product of

Line 268 refer to Table 3 after using the abbreviations Oilconc, protconc, GLS

2.4 put this at least partly to M&M section

Discussion

Line 165 see literature on root growth of rape, e.g. by Eusun Han

https://www.researchgate.net/profile/Eusun_Han and others

2.2 and Discussion

I am aware of the problems with measuring root growth in the field and I like the idea that you clearly mention it  (lines 153-158).

I miss the number of replicates (number of tubes)

I miss a review on literature regarding the effects of N and N imitation on crops (or on oilseed rape)  in the field. What is your hypothesis on this? Will root growth (root length density, rooting depth, …) decrease or increase with N fertilization?

Maybe there is no effect of N fertilization on rooting depth (in case the initial soil N was not too low)?  What did others find regarding this issue?

Author Response

Thank you very much for your constructive review. Please find our responses in attachment.

Best regards.
